# Tumor Necrosis with Adjunction of Preoperative Monocyte-to-Lymphocyte Ratio as a New Risk Stratification Marker Can Independently Predict Poor Outcomes in Upper Tract Urothelial Carcinoma

**DOI:** 10.3390/jcm10132983

**Published:** 2021-07-03

**Authors:** Kun-Che Lin, Hau-Chern Jan, Che-Yuan Hu, Yin-Chien Ou, Yao-Lin Kao, Wen-Horng Yang, Chien-Hui Ou

**Affiliations:** 1Department of Urology, National Cheng Kung University Hospital, College of Medicine, National Cheng Kung University, Tainan 704, Taiwan; typemoondenike@gmail.com (K.-C.L.); greatoldhu@gmail.com (C.-Y.H.); i54921051@gmail.com (Y.-C.O.); pleasewaitforg@hotmail.com (Y.-L.K.); whyang@mail.ncku.edu.tw (W.-H.Y.); 2Division of Urology, Department of Surgery, National Cheng Kung University Hospital Dou-Liou Branch, Yunlin 640, Taiwan; jan.hauchern@gmail.com; 3Institute of Clinical Medicine, College of Medicine, National Cheng Kung University, Tainan 704, Taiwan; 4Department of Urology, College of Medicine, National Cheng Kung University, Tainan 704, Taiwan

**Keywords:** tumor necrosis, monocyte-to-lymphocyte ratio, MLR, upper tract urothelial carcinoma, outcomes

## Abstract

Objectives: This study aimed at investigating the prognostic impact of tumor necrosis and preoperative monocyte-to-lymphocyte ratio (MLR) in patients treated with radical nephroureterectomy (RNU) for upper tract urothelial carcinoma (UTUC). Methods: A total of 521 patients with UTUC treated with RNU from January 2008 to June 2019 at our institution were enrolled. Histological tumor necrosis was defined as the presence of microscopic coagulative necrosis. The optimal value of MLR was determined as 0.4 by receiver operating characteristic (ROC) analysis based on cancer-specific mortality. The Kaplan–Meier method with log-rank test and Cox proportional hazards regression models were performed to evaluate the impact of tumor necrosis and MLR on overall (OS), cancer-specific (CSS), and recurrence-free survival (RFS). Furthermore, ROC analysis was used to estimate the predictive ability of potential prognostic factors for oncological outcomes. Results: Tumor necrosis was present in 106 patients (20%), which was significantly associated with tumor location, high pathological tumor stage, lymph node metastasis, high tumor grade, lymphovascular invasion, tumor size, and increased monocyte counts. On multivariate analysis, the combination of tumor necrosis and preoperative MLR was an independent prognosticator of OS, CSS, and RFS (all *p* < 0.05). Moreover, ROC analyses revealed the predictive accuracy of a combination of tumor necrosis and preoperative MLR for OS, CSS, and RFS with the area under the ROC curve of 0.745, 0.810, and 0.782, respectively (all *p* < 0.001). Conclusions: The combination of tumor necrosis and preoperative MLR can be used as an independent prognosticator in patients with UTUC after RNU. The identification of this combination could help physicians to recognize high-risk patients with unfavorable outcomes and devise more appropriate postoperative treatment plans.

## 1. Introduction

Urothelial carcinomas (UCs) are the fourth most common solid tumor [1]. Upper tract urothelial carcinoma (UTUC) is a rare disease that only accounts for 5–10% of UCs [2]. In patients with localized UTUC, radical nephroureterectomy (RNU) has been the gold-standard treatment until now. However, those with clinically non-metastatic UTUC still experience a high disease recurrence rate and even die from distant metastasis later [3]. Except for tumor stage and lymph node involvement, other pathologically prognostic factors, such as lymphovascular invasion (LVI), tumor architecture, tumor necrosis, tumor size, and concomitant carcinoma in situ, have been proven to be associated with oncological outcomes, which could assist physicians in the clinical decision-making process [4]. Although such prognostic factors have been reported in many previous retrospective studies, most of them lack stronger evidence and information and thus it is necessary to further evaluate their prognostic role to improve our ability to predict oncological outcomes after RNU.

Previously, some studies reported that the presence of extensive tumor necrosis (>10% of the tumor area) is associated with advanced tumor stage and could serve as an independent predictor of worse cancer-specific survival (CSS) and recurrence-free survival (RFS) in UTUC [5,6,7]. At present, most experts believe that tumor necrosis is involved in the cancer-related inflammation response and further promotes tumor growth and progression, thus leading to a poor oncological outcome [8,9]. Moreover, immune cells, such as macrophages, are mediated to accumulate in the hypoxic/necrotic areas of tumors and to participate in tumor growth and metastasis [10]. Recently, an increasing number of articles have reported that the preoperative monocyte-to-lymphocyte ratio (MLR) may be used as a surrogate of cancer-associated inflammation, and high values have a significant correlation with poor prognoses among patients with various types of cancer, including urothelial carcinoma [11,12,13,14,15].

To the best of our knowledge, no studies have provided information about the impact of tumor necrosis in tandem with preoperative MLR on oncologic outcomes in UTUC. Therefore, this study will evaluate whether the presence of tumor necrosis in tandem with high preoperative MLR values contributes to more unfavorable outcomes as compared to the presence of tumor necrosis alone.

## 2. Patients and Methods

### 2.1. Patient Selection

This study was a single-center retrospective review of patients with UTUC who underwent RNU from January 2008 to June 2019. A total of 521 patients were enrolled in this study. Laparoscopic RNU was performed within 30 days of diagnosis, including imaging on computed tomography, computed tomography urography or magnetic resonance urography, high-grade malignancy on endoscopic biopsies, or non-endoscopic manageable tumors, such as multifocal tumors. Lymph node (LN) dissection was performed at the time in the presence of highly suspected LN on preoperative imaging or palpable LNs upon manual retraction of dissected kidney from Gibson incision wound. If patients were diagnosed with bulky advanced UTUC that was not laparoscopically manageable, they were initially arranged to receive neoadjuvant chemotherapy and later considered for surgical intervention.

We recorded the following pathological and clinical features: age, gender, end-stage renal disease under hemodialysis, comorbidities, clinical symptoms, tumor stage (American Joint Committee on Cancer TNM Classification, 7th edition), LN metastasis, tumor location (either ureter or renal pelvis), tumor size, tumor necrosis, pathological grading (2004 WHO classification), lymphovascular invasion (LVI), and preoperative monocyte to lymphocyte ratio (complete blood cell count parameter). Preoperative complete blood counts and differential counts were obtained within 30 days before surgery. As for postoperative pathological examination, hematoxylin-and-eosin-stained slides from routine formalin-fixed and paraffin-embedded specimens were independently re-evaluated by more than two genitourinary pathologists who were blinded to regional lymph node status and clinical follow-up. Tumor necrosis was defined as the presence of microscopic coagulative necrosis, whereas gross-viewed necrosis was not considered histological necrosis, based on the histologic evaluation of all available tumor blocks [5]. The optimal cutoff value for preoperative MLR was 0.4, which was determined by receiver operating characteristic (ROC) curve analysis and the Youden’s index (Appendix A). The exclusion criteria were fever, receiving pre- or perioperative chemo/immunotherapies, lack of intact preoperative serum blood count data, the concurrence of secondary malignancy, autoimmune disease, or chronic systemic inflammation status.

After RNU, the patients were receiving follow-up every three months during the first year, every six months during the second and third year, and then annually. Cystoscopy was performed during every clinical follow-up. Physical examination, history taking, urine analysis, urine cytology, abdominal echo, and abdominal computer tomography were performed as well.

Clinical outcome was managed as follows: overall survival (OS), cancer-specific survival (CSS), and recurrence-free survival (RFS). The OS was the time from RNU till death. The CSS was defined as the time from RNU till death due to UTUC. The RFS was the time from surgery till local or distant recurrence, which did not include metachronous bladder carcinoma.

### 2.2. Statistical Analysis

ROC curve analysis and Youden’s index were performed to determine the optimal cutoff values of preoperative MLR in the prediction of survival outcomes. Kaplan–Meier analysis was used to evaluate OS, CSS, and RFS, and significant differences were determined through the log-rank test. Univariate and multivariate analyses (Cox proportional hazard regression model) were performed to assess prognostic factors. All analyses relied on SPSS 22.0 (SPSS Inc. Chicago, IL, USA) or MedCalc 19.7.2., and all *p* values were two-sided, with <0.05 considered significant. Furthermore, the area under the ROC curve (AUC) was used to calculate the predictive value when the risk factors that we focused on were combined with a set of relevant prognostic factors constituted by pT stage, tumor grade, and LN involvement.

## 3. Result

### 3.1. Associated of Tumor Necrosis with Clinical and Pathologic Characteristics

The correlation between clinical characteristics and tumor necrosis is listed in Table 1. A total of 521 patients were included. Of them, 106 (20%) presented with tumor necrosis, while 415 (80%) did not. The mean age when receiving surgery was 70.7 and 68.7 years in the patient groups with and without tumor necrosis, respectively. The mean follow-up duration was 45.2 months in patients with tumor necrosis and 50.1 months in those without tumor necrosis. When comparing the pathological characteristics between the two groups, the presence of tumor necrosis was significantly associated with tumor location (*p* < 0.001), pT stage (*p* < 0.001), LN invasion (*p* < 0.001), tumor grade (*p* = 0.032), tumor size (*p* < 0.001), LVI (*p* < 0.001), adjuvant chemotherapy (*p* = 0.008), and serum monocyte counts (*p* = 0.004). Furthermore, tumor necrosis had a trend of high-level MLR (*p* = 0.064). The mean value of preoperative MLR was significantly higher in patients with tumor necrosis than in those without tumor necrosis (*p* < 0.05) (Appendix A).

On the contrary, there was no significant difference in age, gender, hemodialysis, diabetes mellitus or hypertension, previous or concomitant bladder cancer history, or preoperative lymphocyte counts (all *p* > 0.05).

### 3.2. Association of Survival with Tumor Necrosis and Preoperative MLR

Our study revealed that tumor necrosis had a significant association with a more inferior OS, CSS, and RFS on Kaplan–Meier analysis (*p* < 0.05) (Figure 1).

In addition, Kaplan–Meier analysis showed that high-level MLR significantly correlated with worse oncological outcomes (Appendix A). Subsequently, we further stratified our patients into three groups based on tumor necrosis and preoperative MLR, including (1) the simultaneous presence of tumor necrosis and high-level MLR (*p* > 0.4) as the high-risk group, (2) tumor necrosis or high-level MLR alone as the intermediate-risk group, and (3) the absence of tumor necrosis and low-level MLR as the low-risk group. Kaplan–Meier analysis showed that the concurrence of tumor necrosis and high-level MLR was significantly associated with decreased OS, CSS, and RFS, compared to the intermediate- and low-risk groups (*p* < 0.05) (Figure 2).

### 3.3. Combination of Tumor Necrosis and Preoperative MLR as an Independent Factor for Predicting Survival

Univariate regression analysis showed that tumor multifocality, tumor location, pT stage, LN metastasis, LVI, tumor size, and the combination of tumor necrosis and preoperative MLR were associated with the prognosis (included OS, CSS, and RFS) of patients with UTUC (all *p* < 0.05; Table 2). Multivariate regression analysis showed that pT stage, LN metastasis, LVI, and concurrence of tumor necrosis and preoperative high-level MLR were also independent prognosticators for a worse OS, CSS, and RFS in patients with UTUC (all *p* < 0.05; Table 2). Next, we further used ROC analysis to examine the predictive ability of a combination of tumor necrosis and preoperative MLR for OS, CSS, and RFS in UTUCs, compared with the basal model (constituted of known prognostic factors: pT stage, tumor grade, and LN involvement). As a result, the addition of a combination of tumor necrosis and preoperative MLR showed that the AUC was 0.745, 0.810, and 0.782, respectively, in predicting OS, CSS, and RFS (all *p*  <  0.001), while the AUC in the basal model was 0.704, 0.781, and 0.768, respectively, for the prediction of OS, CSS, and RFS (Figure 3). Comparing the predictive ability between the two models, adding tumor necrosis and MLR to the basal model was significantly superior to the basal model (*p* < 0.05).

Our study demonstrated that combined use of tumor necrosis and preoperative MLR could be considered a new risk stratification tool to recognize high-risk UTUC patients with unfavorable outcomes after RNU.

## 4. Discussion

The independent prognostic value of tumor necrosis in patients with surgically treated UTUC remains undetermined. It is reasonable that previous studies found inconsistent conclusions since only a few retrospective studies have been conducted with a limited sample size. Moreover, the process of tumor necrosis has been thought to involve cancer-related immune inflammation based on the understanding of the tumor microenvironment. Hence, our study investigated whether tumor necrosis with the adjunction of serum immune inflammation marker MLR may have greater potential in predicting survival in UTUC. Indeed, we discovered that combining tumor necrosis and preoperative MLR was an independent factor for predicting poor oncological outcomes in patients with UTUC after RNU. The addition of preoperative MLR to tumor necrosis was demonstrated to have a better ability to predict oncological outcomes as compared to tumor necrosis alone. Therefore, combining tumor necrosis and preoperative MLR can serve as a new risk stratification marker to guide physicians’ postoperative treatment decision-making.

Previous articles have reported that the presence of tumor necrosis was considered an aggressive behavior of cancer and associated with poor prognosis in many solid malignancies, including renal, prostate, bladder, liver, and colon–rectum [16,17]. In terms of UTUC, several studies revealed that tumor necrosis was significantly associated with poor outcomes of patients after RNU as well [5,6,7]. Lee et al. even demonstrated that tumor necrosis could act as an independent factor for predicting poor survival in 119 patients with UTUC [18]. Our cohort study confirmed that tumor necrosis had a significant correlation with a poor OS, CSS, and RFS by Kaplan–Meier analyses, but we did not find that it alone can independently predict oncological outcomes in multivariate analyses.

The interplay between inflammation and necrosis can contribute to tumor aggressiveness [19]. Macrophage infiltration was found to correlate with tumor necrosis [10,20] and to participate in tumor progression [21,22]. Accordingly, we reevaluated the clinically prognostic value of tumor necrosis in tandem with preoperative MLR. Initially, all patients with UTUC were stratified into three risk groups based on tumor necrosis and preoperative MLR: (1) the concurrent presence of tumor necrosis and high-level MLR as the high-risk group, (2) the presence of tumor necrosis or high-level MLR alone as the intermediate-risk group, and (3) neither tumor necrosis nor high-level MLR as the low-risk group. As a result, the concurrence of tumor necrosis and high-level MLR led to significantly worse outcomes in patients with UTUC after RNU, as compared to the other two groups.

Notably, the presence of tumor necrosis as an independent prognosticator was still controversial until now, while our study demonstrated the value of tumor necrosis accompanying a high MLR value as an independent factor for predicting overall and cancer-specific death, as well as disease progression, in multivariate analyses. In other words, the combination of tumor necrosis and preoperative MLR was superior to tumor necrosis or preoperative MLR alone to predict oncological outcomes of patients with UTUC after RNU. Moreover, the results of the present study suggest that incorporating the combination marker, tumor necrosis and MLR, into clinical practice could help in better predicting prognosis and making decisions regarding further therapies for patients with poor survival outcomes. Tumor necrosis and MLR provided additional predictive value by significantly increasing the index from 0.704, 0.781, and 0.768 to 0.745, 0.810, and 0.782 in OS, CSS, and RFS, respectively.

Our cohort study is the first to demonstrate that the combination of tumor necrosis and preoperative MLR as an integrated marker has more potential to predict survival outcomes in patients with UTUC after surgery than tumor necrosis alone. The pathophysiological mechanism behind the association of tumor necrosis accompanying preoperative MLR with poor survival is described below. The initial development of tumor necrosis was thought to be due to the tumor’s rapid proliferation and outgrowing of its vascular supply, where, in turn, a hypoxic microenvironment was being established. At this time, serum circulating immune cells in the blood, such as monocytes, are mediated by tumor-derived chemokines to migrate into the hypoxic/ischemic areas of the primary tumor [23]. In response to the inflammatory tumor microenvironment, tumor-infiltrating monocytes are induced to differentiate into tumor-associated macrophages (TAMs), which are not only involved in tumor cell death, necrotic cell infiltration, and necrotic tissue formation but also support tumor cell survival, local invasion, and dissemination/metastasis [21,22,24,25,26]. These processes adequately illustrate that there is a reciprocal relationship between tumor necrosis and macrophages and their coordination benefits malignant tumor development/growth.

In the past decade, numerous studies have reported that there is a significant correlation between high values of MLR and poor survival rates in several cancer types [11,13,27,28,29]. For UTUC, several articles have reported that a cutoff value of MLR ranging from 0.28 to 0.5, which was proven to be significantly associated with oncologic outcomes [11,15,30,31,32,33]. MLR is derived from the serum distribution of monocyte and lymphocyte counts. In addition to circulating monocytes, lymphocytes in the blood are equally responsible for the cancer-related immune response. It has been described that decreased circulating lymphocytes can reduce the immune response against tumors [34]. Therefore, increased counts of preoperative serum monocytes and/or decreased counts of lymphocytes correspond to a high value of MLR, which reflects a relatively strong pro-tumor and weak anti-tumor immunity. Consistent with Jan et al.’s study [33], our study determined the optimal cutoff value of MLR as 0.4 by ROC analysis based on cancer-specific mortality. MLR > 0.4 was defined as high-level. We observed that a high-level MLR correlated with worse prognosis in the Kaplan–Meier analysis. The value of MLR was significantly higher in patients with the presence of tumor necrosis than those without (Appendix A), which may imply a positive relation between tumor necrosis and preoperative MLR. As tumor necrosis began to develop and evolve, the expression of high-level MLR could facilitate a variety of elements involved in tumor progression, such as lymphangiogenesis, angiogenesis, and the proliferation, invasion, and migration of tumor cells. The presence of tumor necrosis in combination with high MLR values was indicated to be capable of building up the biological aggressiveness of the cancer itself.

Taken together, in our study, survival analyses included OS, CSS, and RFS. We evaluated the impact of the combination of tumor necrosis and MLR on different kinds of survival rates, providing more information to physicians and assisting in risk stratification for appropriate follow-up treatment planning. Furthermore, MLR was obtained by using a common and simple blood test, and the presence of tumor necrosis was determined by pathologists. In real-world practice, tumor necrosis and MLR, which our study provided, can be clinically applied for routine measurement because of their low cost and easy accessibility. In addition, it is reasonable that patients with tumor necrosis and a high MLR value can be considered to investigate the fundamental pathophysiology of the tumor microenvironment involving tumor necrosis and macrophages and to evaluate the potential benefits of postoperative systemic treatment, such as chemotherapy or immunotherapy, in future prospective studies.

The present study had certain limitations. First, the study data represented a retrospective review of findings from a single center. Second, patients who were not treated surgically were not included in the present analysis. However, of 327 patients with muscle-invasive disease receiving RNU, 54% (N = 178) underwent lymphadenectomy. Third, the lack of information on genetic and molecular biomarkers may reduce the present study’s strength. Nonetheless, further studies are necessary to confirm the role of molecular biomarkers as predictors for unfavorable pathological outcomes of UTUCs.

## 5. Conclusions

Our study demonstrated that the combination of tumor necrosis and preoperative MLR is an independent unfavorable risk factor for OS, CSS, and RFS in patients with UTUC. Combining tumor necrosis and preoperative MLR could assist physicians in recognizing high-risk patients with UTUC and devising more appropriate individualized treatments after surgery in clinical practice.

## Figures and Tables

**Figure 1 jcm-10-02983-f001:**
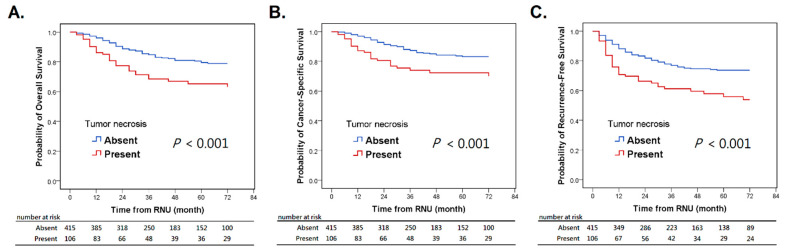
Kaplan–Meier survival curves for overall survival (**A**), cancer-specific survival (**B**), and recurrence-free survival (**C**) in UTUC patients according to tumor necrosis. UTUC: upper tract urothelial carcinoma.

**Figure 2 jcm-10-02983-f002:**
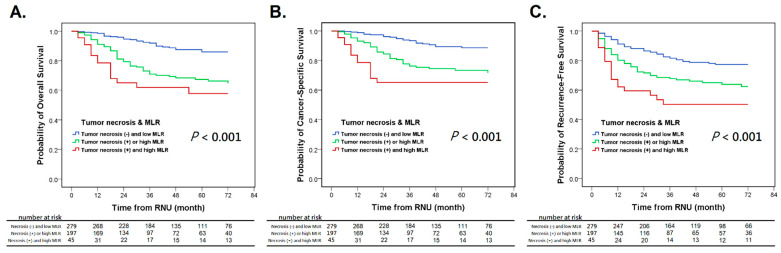
Kaplan–Meier survival curves for overall survival (**A**), cancer-specific survival (**B**), and recurrence-free survival (**C**) in UTUC patients who were divided into three groups based on tumor necrosis and preoperative MLR. UTUC: upper tract urothelial carcinoma, MLR: monocyte-to-lymphocyte ratio.

**Figure 3 jcm-10-02983-f003:**
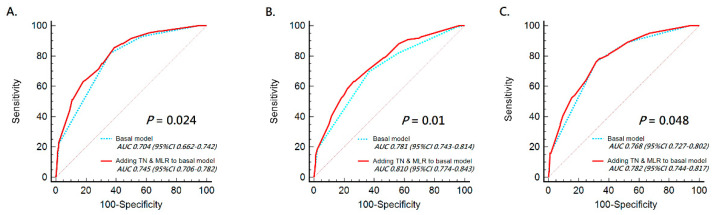
ROC analysis for predictive accuracy of overall survival (**A**), cancer-specific survival (**B**), and recurrence-free survival (**C**) in UTUC patients according to a comparison between the basal model and the basal model combined with tumor necrosis and preoperative MLR. UTUC: upper tract urothelial carcinoma; TN: tumor necrosis; MLR: monocyte-to-lymphocyte ratio.

**Table 1 jcm-10-02983-t001:** Association of tumor necrosis with clinical and pathologic characteristics in patients treated with radical nephroureterectomy for upper tract urothelial carcinoma.

	Total Patients N = 521 (100%)	Tumor Necrosis		
		Absent	Present	
		N = 415 (80%)	N = 106 (20%)	*p* Value
Mean age (year)	69.1 ± 11.2	68.7 ± 11.1	70.7 ± 11.6	
Mean follow-up after surgery (month)	49.1 ± 31.9	50.1 ± 30.6	45.2 ± 36.5	
Age (year)				0.147
≤69	249 (48%)	205 (49%)	44 (42%)	
>69	272 (52%)	210 (51%)	62 (58%)	
Gender				0.601
Male	234 (45%)	184 (44%)	50 (47%)	
Female	287 (55%)	231 (56%)	56 (53%)	
Renal function status				0.098
eGFR ≥ 60 (mL/min/1.73 m^2^)	218 (42%)	182 (44%)	36 (34%)	
eGFR < 60 (mL/min/1.73 m^2^)	214 (41%)	161 (39%)	53 (50%)	
Dialysis	89 (17%)	72 (17%)	17 (16%)	
Hematuria				0.270
No	74 (14%)	48 (13%)	26 (17%)	
Yes	441 (86%)	314 (87%)	127 (83%)	
Hydronephrosis				0.276
No	109 (21%)	72 (20%)	37 (24%)	
Yes	406 (79%)	290 (80%)	116 (76%)	
Prior or concomitant BC				0.861
No	365 (70%)	290 (70%)	75 (71%)	
Yes	156 (30%)	125 (30%)	31 (29%)	
Multifocality				0.417
No	380 (73%)	306 (74%)	74 (70%)	
Yes	141 (27%)	109 (26%)	32 (30%)	
Tumor location				<0.001
Renal pelvis	238 (45%)	173 (42%)	65 (61%)	
Ureter	170 (33%)	157 (38%)	13 (12%)	
Both	113 (22%)	85 (20%)	28 (27%)	
pT stage				<0.001
Tis/Ta/T1	194 (37%)	172 (41%)	22 (21%)	
T2	101 (20%)	85 (21%)	16 (15%)	
T3/T4	226 (43%)	158 (38%)	68 (64%)	
pN stage				0.001
pN0	150 (29%)	108 (26%)	42 (40%)	
pNx	342 (65%)	291 (70%)	51 (48%)	
pN+	29 (6%)	16 (4%)	13 (12%)	
Tumor grade				0.032
Low	26 (5%)	25 (6%)	1 (1%)	
High	495 (95%)	390 (94%)	105 (99%)	
Lyphovascular invasion				<0.001
Absent	370 (71%)	322 (78%)	48 (45%)	
Present	151 (29%)	93 (22%)	58 (55%)	
Tumor size				<0.001
>3 cm	291 (56%)	261 (63%)	30 (28%)	
<3 cm	230 (44%)	154 (37%)	76 (72%)	
Adjuvant chemotherapy				0.008
No	476 (91%)	386 (93%)	90 (85%)	
Yes	45 (9%)	29 (7%)	16 (15%)	
MLR				0.064
Low (≤0.4)	335 (64%)	275 (66%)	60 (57%)	
High (>0.4)	186 (36%)	140 (34%)	46 (43%)	
Monocyte count (10^3^/L)	0.598 ± 0.276	0.551 ± 0.266	0.637 ± 0.304	0.004
Lymphocyte count (10^3^/L)	0.162 ± 0.089	0.162 ± 0.095	0.162 ± 0.062	0.975

RNU = radical nephroureterectomy; eGFR = estimated glomerular filtration rate; BC = bladder cancer; MLR = monocyte-to-lymphocyte ratio.

**Table 2 jcm-10-02983-t002:** Univariate and multivariate Cox regression analyses for predicting overall survival, cancer-specific survival, and recurrence-free survival in patients treated with radical nephroureterectomy for upper tract urothelial carcinoma.

	Overall Survival				Cancer-Specific Survival				Recurrence-Free Survival			
	Univariate		Multivariate		Univariate		Multivariate		Univariate		Multivariate	
	HR (95% CI)	*p*	HR (95% CI)	*p*	HR (95% CI)	*p*	HR (95% CI)	*p*	HR (95% CI)	*p*	HR (95% CI)	*p*
Age at RNU												
>69 yr vs. ≤69 yr	1.264 (0.867–1.844)	0.223	1.599 (0.795–1.808)	0.387	1.409 (0.914–2.172)	0.120	1.359 (0.849–2.174)	0.212	0.959 (0.688–1.338)	0.807	1.009 (0.703–1.448)	0.961
Gender												
female vs. male	0.806 (0.553–1.173)	0.259	0.856 (0.583–1.256)	0.356	0.719 (0.469–1.104)	0.131	0.807 (0.520–1.251)	0.337	0.830 (0.595–1.157)	0.272	0.942 (0.669–1.328)	0.942
Renal function status		0.321		0.960		0.314		0.685		0.425		0.835
eGFR < 60 vs. eGFR ≥ 60	1.377 (0.906–2.093)		0.994 (0.579–2.014)		1.194 (0.756–1.885)		0.856 (0.503–1.459)		0.973 (0.680–1.392)		0.921 (0.618–1.374)	
dialysis vs. eGFR ≥ 60	1.250 (0.729–2.144)		1.080 (0.620–1.490)		0.704 (0.349–1.422)		0.718 (0.327–1.577)		0.716 (0.428–1.197)		0.846 (0.473–1.515)	
Hematuria												
yes vs. no	1.057 (0.613–1.825)	0.841	–	–	0.825 (0.465–1.464)	0.511	–	–	0.845 (0.536–1.332)	0.467	–	–
Hydronephrosis												
yes vs. no	1.017 (0.593–1.490)	0.792	–	–	0.879 (0.527–1.467)	0.623	–	–	0.864 (0.579–1.288)	0.473	–	–
Prior or concomitant BC												
yes vs. no	1.226 (0.827–1.818)	0.311	0.961 (0.620–1.490)	0.859	1.084 (0.686–1.713)	0.731	0.945 (0.565–1.580)	0.829	1.218 (0.858–1.727)	0.270	1.227 (0.839–1.793)	0.291
Multifocality												
yes vs. no	2.610 (1.790–3.806)	<0.001	1.599 (0.694–3.688)	0.271	2.406 (1.56–3.703)	<0.001	1.230 (0.455–3.327)	0.759	1.932 (1.373–2.717)	<0.001	1.251 (0.636–2.464)	0.516
Tumor location		<0.001		0.041		<0.001		0.046		<0.001		0.144
ureter vs. renal pelvis	1.244 (0.761–2.033)		1.984 (1.115–3.424)		1.104 (0.631–1.929)		2.046 (1.091–3.836)		1.130 (0.749–1.705)		1.557 (0.959–2.353)	
both vs. renal pelvis	3.219 (2.061–5.027)		2.060 (0.819–5.185)		2.922 (1.772–4.821)		2.434 (0.826–7.178)		2.245 (1.510–3.338)		2.546 (0.789–3.271)	
pT stage		<0.001		0.004		<0.001		<0.001		<0.001		<0.001
T2 vs. Tis/a/1	1.502 (0.772–2.921)		0.906 (0.430–1.138)		2.978 (1.082–8.194)		1.691 (0.577–4.954)		2.418 (1.229–4.759)		1.667 (0.808–3.442)	
T3/4 vs. Tis/a/1	4.165 (2.515–6.899)		2.386 (1.296–4.394)		12.013 (5.212–27.690)		5.109 (2.009–12.993)		8.359 (4.863–14.370)		4.096 (2.207–7.601)	
pN stage		<0.001		0.001		<0.001		0.001		<0.001		0.001
pNx vs. pN0	0.491 (0.323–0.746)		0.699 (0.430–1.138)		0.322 (0.198–0.526)		0.610 (0.353–1.053)		0.413 (0.287–0.595)		0.705 (0.467–1.066)	
pN+ vs. pNx/0	4.047 (2.360–6.939)		2.386 (1.296–4.394)		4.312 (2.467–7.535)		2.890 (1.593–5.243)		4.130 (2.528–6.748)		2.087 (1.197–3.638)	
Tumor grade												
high vs. low	2.833 (0.699–11.480)	0.145	–	–	4.390 (0.611–31.536)	0.141	–	–	7.775 (0.987–55.594)	0.051	–	–
Lymphovascular invasion												
present vs. absent	3.059 (2.099–4.456)	<0.001	1.684 (1.073–2.645)	0.024	4.627 (2.984–7.176)	<0.001	2.047 (1.227–3.415)	0.006	4.512 (3.219–6.323)	<0.001	2.449 (1.665–3.601)	<0.001
Tumor size												
>3 cm vs. ≤3 cm	1.699 (1.164–2.478)	0.006	0.900 (0.575–1.409)	0.645	2.125 (1.371–3.293)	0.001	0.988 (0.588–1.661)	0.965	1.751 (1.253–2.447)	0.001	0.980 (0.668–1.439)	0.919
Adjuvant chemotherapy												
yes vs. no	1.457 (0.799–2.657)	0.219	0.506 (0.246–1.042)	0.064	1.931 (1.048–3.561)	0.035	0.521 (0.249–1.092)	0.084	2.963 (1.907–4.605)	<0.001	0.925 (0.545–1.570)	0.772
TN and MLR		<0.001		<0.001		<0.001		0.009		<0.001		0.105
TN (+) or high MLR vs. TN (–) and low MLR	2.87 (1.875–4.413)	<0.001	2.009 (1.246–3.240)	0.004	2.907 (1.779–4.751)	<0.001	1.741 (1.001–3.032)	0.050	1.973 (1.375–2.831)	<0.001	1.283 (0.859–1.917)	0.223
TN (+) and high MLR vs. TN (–) and low MLR	4.776 (2.687–8.492)	<0.001	3.784 (1.943–7.370)	<0.001	5.341 (2.813–10.142)	<0.001	3.233 (1.513–6.907)	0.002	3.294 (1.970–5.508)	<0.001	1.867 (1.011–3.340)	0.035

RNU = radical nephroureterectomy; eGFR = estimated glomerular filtration rate; BC = bladder cancer; TN = tumor necrosis; MLR = monocyte-to-lymphocyte ratio.

## Data Availability

The data presented in this current study are available on reasonable request from the corresponding author. The data are not publicly available due to the recommendations of the ethics committee.

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
