# Peer review of "Tumor Necrosis with Adjunction of Preoperative Monocyte-to-Lymphocyte Ratio as a New Risk Stratification Marker Can Independently Predict Poor Outcomes in Upper Tract Urothelial Carcinoma"

_jcm, 2021, doi:10.3390/jcm10132983_

Round 1
Reviewer 1 Report
Upper urinary tract cancer (UTUC) management is significantly changing, and some landmark studies are suggesting that recognizing high-risk patients, candidate for adjuvant treatments, could improve survival. In this context, this study is definitely interesting.
I would advise some improvements:
1) In the "Patients selection" section, it would be useful to clarify the criteria that brought patients to RNU: imaging only, high grade TCC at endoscopic biopsies, not-endoscopically manageable TCC etc;
2) In the same section, the Authors stated that all patients underwent a laparoscopic procedure: this choice should be discussed, especially regarding bulky disease and lymph nodes involvement.
3) 63% of patients had pT2-4 disease, but only 9% received adjuvant chemotherapy. Is it due to the fact that this cohort of patients was treated before the POUT study? This should be clarified.
Author Response
Dear Editor and Reviewers,
We would like to thank the Editorial Board of the Journal of Clinical Medicine for their consideration of our manuscript “Tumor necrosis with adjunction of preoperative monocyte-to-lymphocyte ratio as a new risk stratification marker can independently predict poor outcomes in upper tract urothelial carcinoma” (JCM-1226766) and we would also like to thank the reviewers for their thoughtful comments and helpful suggestions.
We believe that we have satisfactorily addressed the comments of the reviewers, and that addressing these points has indeed significantly improved our manuscript. Below we have provided a point by point response to all the reviewer’s comments.
(Reviewer #1):
- The reviewer’s comment: In the "Patients selection" section, it would be useful to clarify the criteria that brought patients to RNU: imaging only, high grade TCC at endoscopic biopsies, not-endoscopically manageable TCC etc.
In response to the reviewer’s suggestion, we inserted and rephrased “Laparoscopic RNU was performed within 30 days from diagnosis, including imaging on computed tomography, computed tomography urography or magnetic resonance urography, or high grade malignancy on endoscopic biopsies, or non-endoscopical manageable tumors, such as multifocal tumors.” in the “Patients and methods” section (Page 2, lines 76-79)
- The reviewer’s comment: In the same section, the Authors stated that all patients underwent a laparoscopic procedure: this choice should be discussed, especially regarding bulky disease and lymph nodes involvement.
To address the reviewer’s concern, patients with bulky advanced UTUC initially received neo-adjuvant chemotherapy and later considered surgical intervention. In our study, patients treated with neoadjuvant chemotherapy had been omitted. All enrolled patients belonged to laparoscopic manageable UTUC. Given the improvement of techniques and surgeons’ experience, the criteria of LNU have been dramatically expanded. Patients with advanced stage UTUC also underwent LNU, resulting in similar oncological outcomes as open method. All cases in our study underwent laparoscopic radical nephroureterectomy with bladder cuff excision. In our study, no case was operated in open RNU.
Hence, we inserted and rephrased “Lymph node (LN) dissection would be performed at the time if presence of highly suspected LN on pre-operative imaging or palpable LNs upon manual retraction of dissected kidney from Gibson incision wound. If patients were diagnosed of bulky advanced UTUC which was not laparoscopically manageable, they would be initially arranged to receive neoadjuvant chemotherapy and later considered surgical intervention.” in the “Patients and methods” section (Page 2, lines 79-84).
- The reviewer’s comment: 63% of patients had pT2-4 disease, but only 9% received adjuvant chemotherapy. Is it due to the fact that this cohort of patients was treated before the POUT study? This should be clarified.
To address the reviewer’s concern, our patients were collected from 2008 to 2019. Less than 10% patients who were diagnosed of localized advanced UTUC were allowed to receive adjuvant chemotherapy because there was no strong evidence regarding benefit of adjuvant systemic chemotherapy for localized advanced UTUC after surgery before the POUT trial.
In conclusion, we again thank the reviewers of our manuscript for their helpful comments which we believe have substantially improved our manuscript. We hope that the reviewers will now find our manuscript suitable for publication in JCM, and eagerly await the response to our revisions.

Reviewer 2 Report
This single center, retrospective study examined the risk factors of non-metastatic upper tract urothelial (UTUC).
The authors found Tumor necrosis with adjunction of preoperative monocyte-to-lymphocyte ratio can predict UTUC with poor prognosis.
I think this is a serious problem because there is currently no clinically applicable risk category for UTUC. In this context, this report is interesting.
However, several issues need to be addressed:
Major
- The MLR cutoff value of 0.4 was determined in your study, but how does this value compare to previous studies? Is it reasonable? Please discuss this point.
- The authors presented a new risk-stratification tool to recognize UTUC with poor prognosis. However, the AUC seems to be almost the same as the basal model. Furthermore, we thought that the basal model has the advantage that the parameters are more readily available. In this context, the implications of the authors' model should be discussed.
- The authors described this new risk-stratification tool as being informative for decision-making in postoperative treatment. The authors should discuss how to use this model in real world clinical practice.
Minor
- (Abstract) I think the description of the method is poor. The author should add the study design, and the definition and explanation of the key word “tumor necrosis”.
- Is the description of gender and sex a misprint? (Line 75)
- You have excluded patients with fever, but did you also exclude patients with tumor fever?
- The authors should describe how to diagnose tumor necrosis in the Method section. If there is a reference to that method, the authors should cite it.
- Who performed the pathological diagnosis? This should be stated.
- When was the blood sample taken to calculate the MLR? This point should be stated.
Author Response
Dear Editor and Reviewers,
We would like to thank the Editorial Board of the Journal of Clinical Medicine for their consideration of our manuscript “Tumor necrosis with adjunction of preoperative monocyte-to-lymphocyte ratio as a new risk stratification marker can independently predict poor outcomes in upper tract urothelial carcinoma” (JCM-1226766) and we would also like to thank the reviewers for their thoughtful comments and helpful suggestions.
We believe that we have satisfactorily addressed the comments of the reviewers, and that addressing these points has indeed significantly improved our manuscript. Below we have provided a point by point response to all the reviewer’s comments.
(Reviewer #2)
Major
- The reviewer’s comment: The MLR cutoff value of 0.4 was determined in your study, but how does this value compare to previous studies? Is it reasonable? Please discuss this point.
To address Reviewer’s concern, according to previous studies of upper tract urothelial carcinoma, LMR, of which optimal value ranged from 2.0 to 3.6, was demonstrated to be associated with oncological outcomes after surgery (DOI:10.1038/s41598-019-42781-y;DOI: 10.22037/uj.v0i0.4120; DOI:10.1136/jclinpath-2014-202658; DOI: 10.2147/OTT.S97520; DOI :10.1007/s10147-017-1150-x). If LMR mathematically conversed to MLR, its value was from 0.28 to 0.5. Jan et al. had reported that MLR’s value was 0.4 and significantly associated with survival outcomes. In this present study, it was reasonable and acceptable that MLR value was defined as 0.4, which was also the same as Jan et al. study (DOI:10.1245/s10434-018-6942-3).
Additionally, we inserted two sentences “For UTUC, several articles had reported that the cutoff value of 258 MLR ranged from 0.28 to 0.5, which was proved to be significantly associated with 259 oncologic outcomes” (Page 8, lines 258-259) and ”Consistent with Jan et al. study, our study determined the optimal cutoff value of MLR as 0.4 by ROC analysis based on cancer-specific mortality. MLR > 0.4 was defined as high-level”(Page 8, lines 266-268) in the discussion section. We also updated four references (DOI: 10.22037/uj.v0i0.4120; DOI:10.1136/jclinpath-2014-202658; DOI: 10.2147/OTT.S97520; DOI :10.1007/s10147-017-1150-x).
- The reviewer’s comment: The authors presented a new risk-stratification tool to recognize UTUC with poor prognosis. However, the AUC seems to be almost the same as the basal model. Furthermore, we thought that the basal model has the advantage that the parameters are more readily available. In this context, the implications of the authors' model should be discussed.
In response to the reviewer’s suggestions and concerns, the results of the present study suggested that incorporating the combination marker, tumor necrosis and MLR, into the clinical practice could help in better predicting the prognosis and making decisions regarding further therapies for patients with poor survival outcomes. The presence of tumor necrosis as an independent prognosticator is still controversial until now, while tumor necrosis accompanying high MLR value was demonstrated to be an independent factor for predicting OS, CSS, and RFS in our study. Tumor necrosis and MLR provided additional predictive value by increasing the index from 0.703, 0.779, 0.766 to 0.738, 0.802, 0.778 in OS, CSS, and RFS, respectively.
Hence, we rephrased the paragraph as below in the “discussion” section (Page 7, lines 234-239). “ Moreover, the results of the present study suggested that 234 incorporating the combination marker, tumor necrosis and MLR, into the clinical 235 practice could help in better predicting the prognosis and making decisions regarding 236 further therapies for patients with poor survival outcomes. Tumor necrosis and MLR 237 provided additional predictive value by increasing the index from 0.703, 0.779, 0.766 to 238 0.738, 0.802, 0.778 in OS, CSS, and RFS, respectively.”
- The reviewer’s comment: The authors described this new risk-stratification tool as being informative for decision-making in postoperative treatment. The authors should discuss how to use this model in real world clinical practice.
To address Reviewer’s concern, our risk-stratification tool composed of preoperative serum MLR and postoperatively histological tumor necrosis. MLR was obtained by using a common and simple blood test, and presence of tumor necrosis was determined by the pathologists. In the real world practice, tumor necrosis and MLR in our study can be clinically applied for routine measurement because of their low cost and easy accessibility. This combination marker has been demonstrated to be an independent prognostic factor and might be useful parameters for further classification of patients who should be considered receiving post-operative systemic treatment. However, MLR might be influenced by several external factors, such as infection, medication, and lifestyle habits. The utility of a blood test for predicting survival should be assessed cautiously.
In the future, we believed that through a well-design prospective study, more patients were required to develop an effective model to obtain more strong evidence of ACT benefit for localized UTUC with tumor necrosis and elevated MLR.
Finally, we inserted “Taken together, in our study, survival analyses included OS, CSS, and RFS. Our study evaluated the impact of combination of tumor necrosis and MLR on different kinds of survival rate, providing more information to physicians and assisting in risk stratification for follow-up appropriate treatment planning. Furthermore, MLR was obtained by using a common and simple blood test, and presence of tumor necrosis was determined by the pathologists. In the real world practice, tumor necrosis and MLR in our study can be clinically applied for routine measurement because of their low cost and easy accessibility. Besides, it is reasonable that patients with tumor necrosis and high MLR value are considered to investigate the fundamental pathophysiology of tumor microenvironment involving tumor necrosis and macrophages and to evaluate the potential benefit from postoperative systemic treatment, such as chemotherapy or immunotherapy, in future prospective studies” in the “discussion” section (Page 8, lines 277-288).
Minor
- The reviewer’s comment: (Abstract) I think the description of the method is poor. The author should add the study design, and the definition and explanation of the key word “tumor necrosis”.
In response to the reviewer’s suggestion, we inserted a sentence” Histological tumor necrosis was defined as the presence of microscopic coagulative necrosis. The optimal value of MLR was determined as 0.4 by receiver operating characteristic (ROC) analysis based on cancer-specific mortality” in the” method” section in the Abstract (Page 1, lines 21-23), and rephrased “Kaplan-Meier method with log-rank test and Cox proportional hazards regression models were performed to evaluate the impact of tumor necrosis and MLR on overall (OS), cancer-specific (CSS), and recurrence-free survival (RFS)” in the” method” section in the Abstract (Page 1, lines 23-25).
- The reviewer’s comment: Is the description of gender and sex a misprint? (Line 75)
In response to the reviewer’s suggestion, the word “gender” has been revised (Page 2, line 85).
- The reviewer’s comment: You have excluded patients with fever, but did you also exclude patients with tumor fever?
To address the reviewer’s concerns, neoplasm fever often occurred in the lymphoma and leukemia patients, and in general, its infection origin was unable to be identified or did not exist. In our study, we excluded UTUC patients with fever episode before surgery who were diagnosed of infected diseases, including cellulitis, upper respiratory tract or urinary tract infection. These patients received antibiotic treatment, and fever/associated symptoms resolved thereafter. We concerned that preoperative complete blood data was influenced by infection-related immune response. Thus, we purposely eliminated the impact of infection factor on serum immune cell distributions as much as possible and avoided such the factor to misinterprint our study data.
- The reviewer’s comment: The authors should describe how to diagnose tumor necrosis in the Method section. If there is a reference to that method, the authors should cite it.
In response to the reviewer’s suggestion, tumor necrosis definition (the presence of microscopic coagulative necrosis, whereas gross-viewed necrosis was not represented histological necrosis, based on the histologic evaluation of all available tumor blocks) has been inserted in the “Patients and methods” section (Page 3, lines 95-98) and also updated one reference (DOI:10.1016/j.juro.2006.04.019).
- The reviewer’s comment: Who performed the pathological diagnosis? This should be stated.
In response to the reviewer’s suggestion, we inserted a sentence “As for postoperatively pathological examination, hematoxylin and eosin stained slides from routinely formalin fixed and paraffin embedded material were independently re-evaluated by more than two genitourinary pathologists who were blinded to regional lymph node status and clinical follow-up” in the “Patient and methods” section (Page 2/3, lines 92-95).
- The reviewer’s comment: When was the blood sample taken to calculate the MLR? This point should be stated.
In response to the reviewer’s suggestion, we inserted a sentence “preoperative complete blood counts and differential counts were obtained within 30 days before surgery” in the “Patient and methods” section (Page 2, lines 90-92).
In conclusion, we again thank the reviewers of our manuscript for their helpful comments which we believe have substantially improved our manuscript. We hope that the reviewers will now find our manuscript suitable for publication in JCM, and eagerly await the response to our revisions.

Round 2
Reviewer 2 Report
The authors responded to the reviewers' comments.
Author Response
Dear Editor and Reviewers,
We appreciate the time and effort that you, the reviewers, and the academic editor have dedicated to providing your valuable feedback on my manuscript. We are grateful to you for your insightful comments on my paper. We have been able to incorporate changes to reflect most of the suggestions provided by you. We have highlighted the changes within the manuscript [file name: jcm-1226766 (revised v2.0)].
